# Experimental research on the impact pressure of tidal bore fronts

**Shubo Yue**[1]*, **Jian Zeng**[1], **Yongping Chen**[2], **Zhiyong Zhang**[1], **Dongfeng Xie**[1]

1 Key Laboratory of Estuary and Coast of Zhejiang Province, Zhejiang Institute of Hydraulic and Estuary, Hangzhou, China, 2 State Key Laboratory of Hydrology-Water Resources and Hydraulic Engineering, Hohai University, Nanjing, China

* yueshubo@sina.com

**Data Availability Statement:** All relevant data are within the manuscript and its Supporting information files.

**Funding:** This study was financially supported by Zhejiang Provincial Natural Science Foundation of

## Abstract

Tidal bore impact can be strong and destructive, placing estuarine infrastructures under great threat. However, there is a lack of research focusing on accurately estimating the impact pressure exerted by tidal bores. Herein new experiments were conducted to investigate the pressure of tidal bore fronts in a glass flume. Through analysis of instantaneous pressure of three forms of tidal bore, it was observed that the pressure fluctuation of weak and strong breaking bore fronts is characterized by impact pressure. The vertical distribution and maximum impact pressure of tidal bore were studied. The maximum impact pressure of breaking bore fronts appeared around 0.46 times height of it. The relationship between relative impact pressure and height of the tidal bore fronts was found to closely follow a normal probability density function. Through nonlinear regression analysis, an empirical equation was derived to calculate impact pressure, which was validated using observation data from the Qiantang River in China. This equation can be utilized to predict the impact pressure of tidal bore fronts and provide valuable support for estuarine engineering design.

## Introduction

A tidal bore is formed when tidal flow travels upstream against the current of a river or narrow estuary. Famous tidal bores can be found in various locations around the world, such as the Qiantang River in China, the Seine River in the UK, the Mont Saint-Michel Bay in France, the Petitcodiac River in Canada and the Colorado River in Mexico. The Qiantang River tidal bore, in particular, is renowned for its grand scale, reaching several meters in height and creating a spectacular display of powerful waves. Tidal bores have a significant effect on the local ecosystems and human activities. However, the immense energy of tidal bores can also cause erosion and damage to river banks and estuarine infrastructures. It is essential to recognize and assess the effect of tidal bores.

Tidal bores are magnificent natural occurrence spectacles that showcases the dynamic interaction between the ocean and river systems. Scholars have systematically observed and studied this phenomenon to gain a better understand. The bore development is closely linked with the tidal range and the boundary conditions of river mouth. Shi et al. [1] investigated how the slope of tidal bores fronts varied with changes in estuariy channel morphology. They

China (LY20E090001), Joint Funds of Zhejiang
Provincial Natural Science Foundation of China and
Water Resources Department
(LZJWD22E090002), Science and Technology
Program of Zhejiang Provincial Department of
Water Resources (RA2210), National Nature
Science Foundation of China(42176170), Key
project of Zhejiang Provincial Natural Science
Foundation (LZJWZ23E090003). The funder had
no role in study design, data collection and
analysis, decision to publish, or preparation of the
manuscript.

**Competing interests:** The authors have declared
that no competing interests exist.

found that decreases in channel width and depth promoted the generation of tidal bores. Lynch [2] argued that a strong breaking bore appears when the ratio of downstream depth to upstream depth is greater than 1.4. Koch and Chanson [3] conducted a flume experiment to study the critical Froude number for the break of the free surface of tidal bore. For surge Froude numbers less than 1.7, the bore was an undular bore. For Fr> 1.7, a breaking bore was observed and the fronts had a marked roller. Based on the form of the tidal bore fronts, tidal bores can be classified into three forms: undular bore, weak breaking bore and strong breaking bore.

The undular bore, characterized by a smooth wave fronts and a train of secondary waves, propagates upstream relatively slowly, with the free-surface undulations having a smooth appearance. Tricker [4] gave an introduction of undular bore. Chen et al. [5] argued the formation condition and initial site of undular bore in the north branch of Yangtze River estuary. Madsen and Svendsen [6] developed a theoretical model for the velocity field and the surface profile of bores and hydraulic jumps. They [7] also discussed the turbulence, which is concentrated in a wedge that originates at the toe of the fronts and spreads towards the bottom. Chanson [8, 9] utilized physical model to study the flow field and the mixing and dispersion of tidal bore. The experimental data highlighted rapid flow redistributions between successive wave troughs and crests as well as large bottom shear stress variations. Wolanskia et al. [10] analyzed the measurements in the macro-tidal Daly Estuary, which showed that the presence of an undular bore contributed negligibly to the dissipation of tidal energy. Simpson et al. [11] analyzed undular bore level, flow velocity, Reynolds shear stress, and turbulent kinetic energy, based on observed data in a tidally energetic estuarine channel of almost uniform cross-section.

Breaking bores, characterized by a single breaking roller followed by a few secondary waves (weak breaking bore) or violent aeration flow (strong breaking bore), exhibit different hydrodynamic characteristics from undular bore. Pierre et al.[12] present numerical simulation of a weak breaking bore, while Leng et.al [13] and Shi et.al [14] studied the roller characteristics of breaking bore, revealing a strongly three-dimensional turbulent flow motion. Pan et al. [15] deduced a relationship between the tidal bore propagation velocity and the height of the upper and ebb stream based on the one-dimensional continuity equation and momentum equation. Huang et al. [16, 17] studied hydrodynamic characteristics such as propagation velocity, height, and vertical distribution of velocity by simulating strong breaking bores in a glass flume. Xie et al. [18, 19] studied the hydrodynamic and turbulence characteristics of the strong breaking tidal bore of the Qiantang River, China, using field-measured data. Yue et al. [20, 21] recorded and presented the aeration process and aeration form of tidal bore in a glass rectangular flume. Meanwhile, they analyzed and discussed the relationship among the intensity of tidal bore, the Froude number of tidal bore and the relative aeration length of tide bore fronts.

It is important to note that the strong breaking bore, particularly the tidal bore in the Qiantang River (China), has a dangerous and notorious reputation. As shown in Fig 1, the severe damage to a pier (in 1988) and a seawall (in 2017) were caused by tidal bores in the Qiantang River.

Understanding its pressure characteristics holds practical significance for engineers and scientists. Xu [22] noted that bore pressures on the sheet-pile groin decreased slightly with the depth and were characterized by a trapezoidal distribution below the still water level. In measurements of bore pressure on piers, Shao et al. [23] observed that the bore fronts pressure sinusoidally varies in the few seconds to few minutes after the first bore wave collides with a structure. Chen et al.[24] conducted field observations of tidal bore pressure on piled spur dikes. They found that the pressure above the level of ebb flow is distributed in a triangular shape, while the pressure below the level of ebb flow is distributed in a parabolic shape. Shen et al. [25] conducted a study on the tidal bore pressure exerted on the Yulin stone pond in the

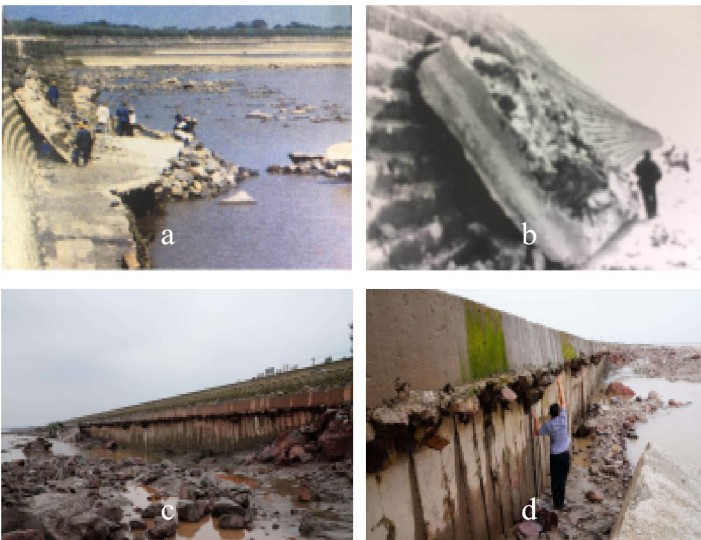

**Fig 1. Photos of the damage situation of a pier (a, b in 1988) and a seawall (c, d in2017) in the Qiantang River.**

Haining section of the northern bank of the Qiantang River, China. The distribution of pressure is parabolic or triangular in shape. Li et al. [26] investigated the time-averaged and turbulence characteristics of tidal bore pressure, and established a fitting relationship between the pressure extremum and the height of the tidal bore, which aligned with an exponential distribution law. Additionally, they investigated the impact of tidal bores on the trestle piers of a river-crossing bridge using field tests. Their also revealed that the maximum dynamic impact pressure on the seaward side of the pier occurs at the base of the bore [27]. Chen [28] estimated that the magnitude of the tidal force is directly proportional to the square of the tidal height by conducting a model experiment on the tidal force acting on the pier body. Khezri and Chanson [29] investigated the primary factor contributing to the inception of sediment transport,which is the longitudinal pressure gradient force of breaking bore. Zhang et al. [30] analyzed the surface pressure and force of piles under a tidal bore using a physical test model. They observed that the maximum pressure of the tidal bore is correlated with the square of the propagation velocity of the tidal bore, but they only derived a maximum pressure calculation formula. However, equations describing how pressures caused by tidal bore impacts have not been presented.

In this study, we conducted experiments by simulating tidal bores in a glass rectangular flume. We obtained the the instantaneous pressure of tidal bore fronts and explored the vertical variation law of the impact pressure of tidal bore fronts.

## Physical model

The glass rectangular flume measures 50 m in length, 1.2 m in width, 0.6 m in height, with a flat bottom. Fig 2 shows the experimental apparatus for generating the tidal bore at one end of the flume. Water was supplied by the submersible pumps to the expectant level in the water intake through the plate of energy dissipation. Tidal bores were generated by rapidly laying the gate plate flat to let the water rush into the downstream against the flow from the other end of the flume. As details about the glass rectangular flume, Yue et al. [20, 21] had given the same setup to generate tidal bores. As shown in Fig 3, three forms of tidal bore in the glass flume were generated, which were consistent with forms of field observations in the Qiantang River in 2010 and 2022.

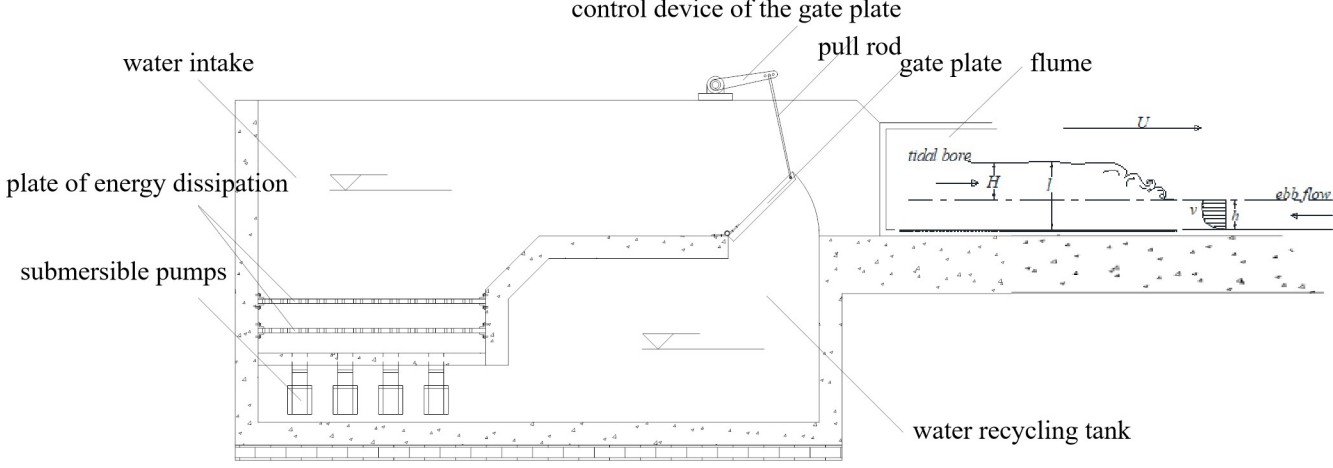

**Fig 2. The experimental apparatus for generating the tidal bore at the downstream end of the flume.**

Tide and runoff are primary factors influencing the magnitude of the tidal bore [3]. The height and propagation velocity of the tidal fronts are tide factors, while depth and velocity of ebb flow are runoff factors. In the experiment, height of tidal bore, as well as depth and velocity of ebb flow, was chosen as controlling parameters because the propagation velocity of the tidal fronts is linked to its height [15]. Three values of 0.042 m, 0.073 m, and 0.095 m were tested to reflect changes in depth of ebb flow (the depth from the water surface of ebb flow to the bottom of bed, represented by $h$). Ebb flow velocities (represented by $v$) were tested at three values: 0 m/s, 0.2 m/s, and 0.3 m/s. Height of tidal bore fronts (the height from the water surface of ebb flow to the water surface of tidal bore fronts, represented by $H$) were tested in 11 experimental values between 0.010 m and 0.20 m. The measurement method for those parameters of tidal bore and ebb flow is shown in Fig 4. Three capacitive wave-height sensors are equidistantly distributed at 2m in length ways ($\Delta X = 2m$). During the bore passage, the second sensor are used to record the depth of ebb flow ($h$) and the level of tidal bore fronts ($l$), but also they are used to obtain the interval times $\Delta t_1$ and $\Delta t_2$ (units of time is second) between the three

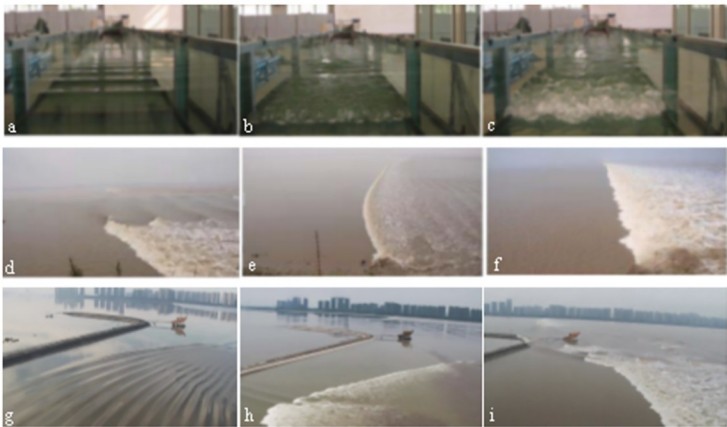

**Fig 3. Three forms of tidal bore generated in the glass flume(a,b,c) and observed in the Qiantang River(d,e,f at Yanguan Station in 2010; g,h,i at Meinvba Station in 2022).**

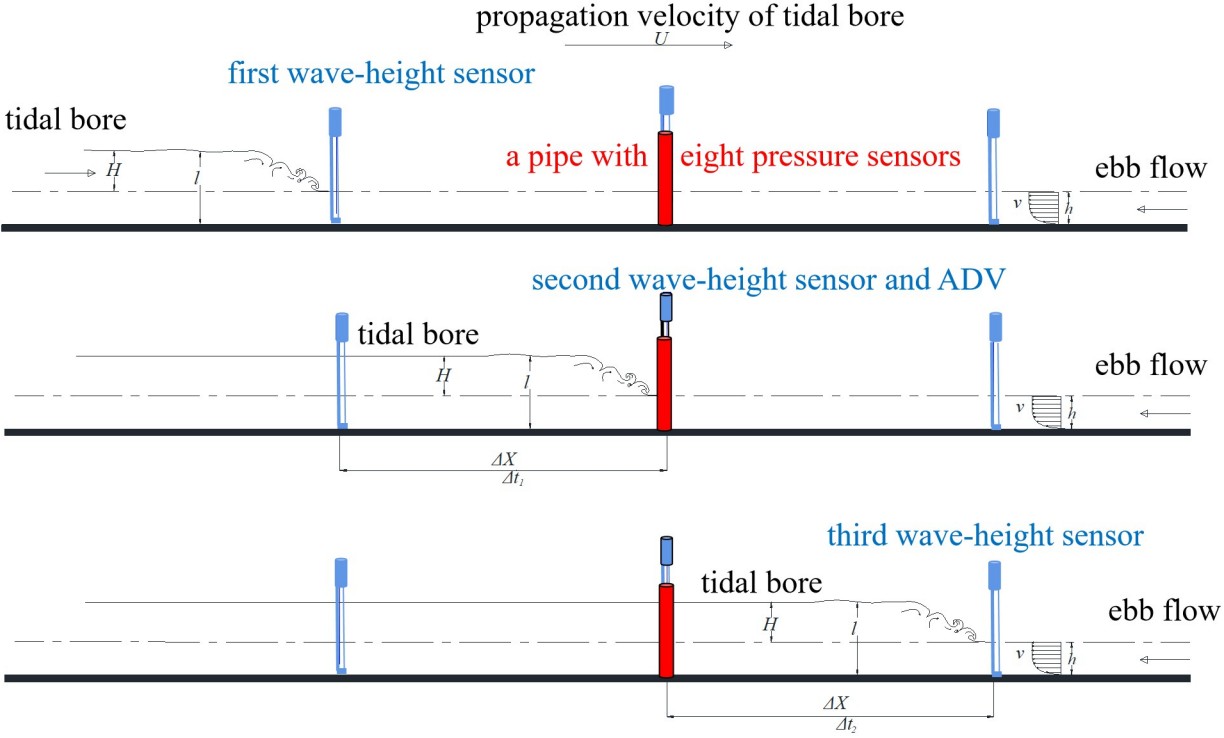

**Fig 4. The measurement method for those parameters of tidal bore and ebb flow.**

sensors. $H$ was calculated by $l - h$, and the mean propagation velocity of tidal bore fronts (represented by $U$) was calculated by $\frac{\Delta X(\Delta t_1 + \Delta t_2)}{2\Delta t_1 \Delta t_2}$.

Meanwhile, positioned parallel to the second wave-height sensor as shown in Fig 5, an Acoustic Doppler Velocimeter (ADV) was used to measure the velocity of ebb flow($v$), while eight thin film piezoelectric pressure sensors are installed in a 0.015 m diameter Poly Vinyl Chloride (PVC) pipe. Sensors are oriented in the direction of the tidal inflow and spaced 0.03 m apart. They are labeled T1 to T8 from bottom to top. Sensors T1 to T3 measure the pressure of ebb flow, while sensors T4 to T8 measure the instantaneous pressure of the tidal bore fronts. Before data collection, zero calibration is performed in the air and pressure is tested and adjusted right in the water. The collected data will undergo further noise reduction processing to obtain pressure of tidal bores. The sampling frequency is 100 Hz.

## Results and discussion

### Form of tidal bore

Through the glass sidewall of the flume, a high-speed camera was used to record instantaneous free-surface profiles of the bore fronts. Fig 6 displays profiles captured by the camera, while Fig 7 shows the tidal levels of the bore fronts collected by the second wave-height sensor when $h = 0.096m$ and $v = 0.2m/s$.

It is shown that the form of the tidal bore is dependent on the height and the mean propagation velocity of the tidal bore fronts when the ebb flow remains constant. This implies that the transformation of the tidal bore form, from an undular bore to a strong breaking bore, depends on the Froude number of the tidal bore ($Fr$) and the relative height of the tidal bore

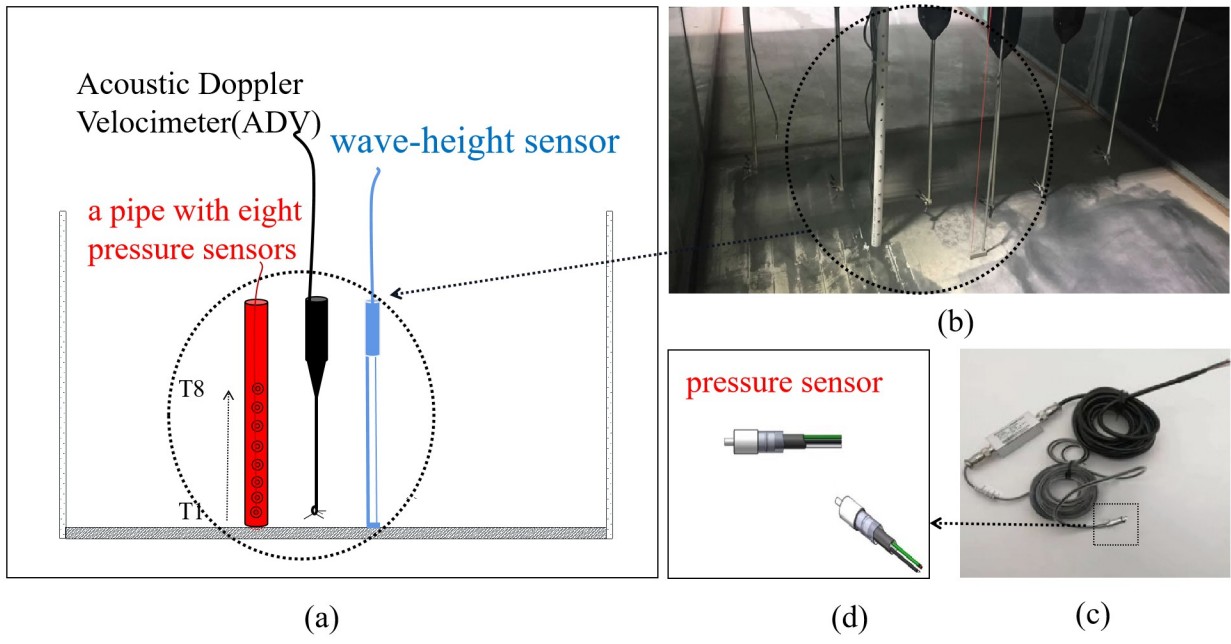

**Fig 5. Schematic diagram and photograph of the experimental setup(a:diagram; b: the photo; c and d: the thin film piezoelectric pressure sensor).**

($H/h$). The Froude number of the tidal bore is calculated by:

$$Fr = \frac{U - v}{\sqrt{gh}} \tag{1}$$

where the propagation velocity of tidal bore fronts ($U$) is positive and the ebb flow($v$) is negative.

   Through experiments in the flume, forms of the tidal bores have been identified and categorized based on by $Fr$ and $H/h$. Fig 8 shows that the three forms of tidal bores vary systematically

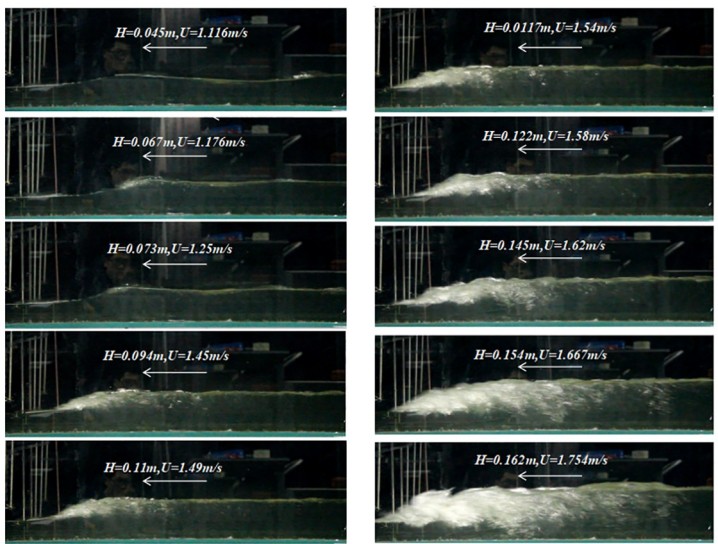

**Fig 6. Profiles of Tidal bore fronts captured by the camera when $h = 0.096m$ and $v = 0.2m/s$.**

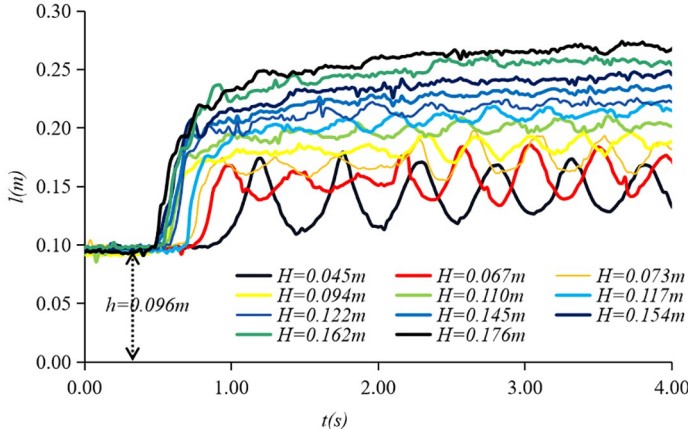

**Fig 7. Tidal levels of the bore fronts collected by the second wave-height sensor when *h* = 0.096*m* and *v* = 0.2*m/s*.**

with *Fr* and *H/h*. When *Fr* < 1.4 and *H/h* < 0.5, the undular bore propagates relatively slowly with the free-surface undulations having a smooth appearance. As 1.4 < *Fr* < 1.7 and 0.5 < *H/h* 003C *0.8*, the weak breaking bore appeared with a single breaking roller followed by a few secondary waves. When *Fr* > 1.7 and *H/h* > 0.8, the strong breaking bore appeared with breaking roller followed by violent aeration flow. The findings from our experiments are largely consistent with conclusions of Koch and Chanson [3]. Xie et al.[18, 19] also observed three forms of tidal bore in the Qiantang River and pointed out the relationship with the Froude number *Fr*. When it exceeds 1.3, the tidal bore surface is no longer smooth. When it is between 1.3 and 1.4, the free surface breaks and forms water roller on the tidal bore fronts. Strong breaking tidal bores occur when it exceeds 1.7. Our results are also consistent with their results of based on field measurements in the Qiantang River. The agreement of the experimental and field observations suggests that the tidal bore in the flume exhibits similar characteristics to those observed in natural environments.

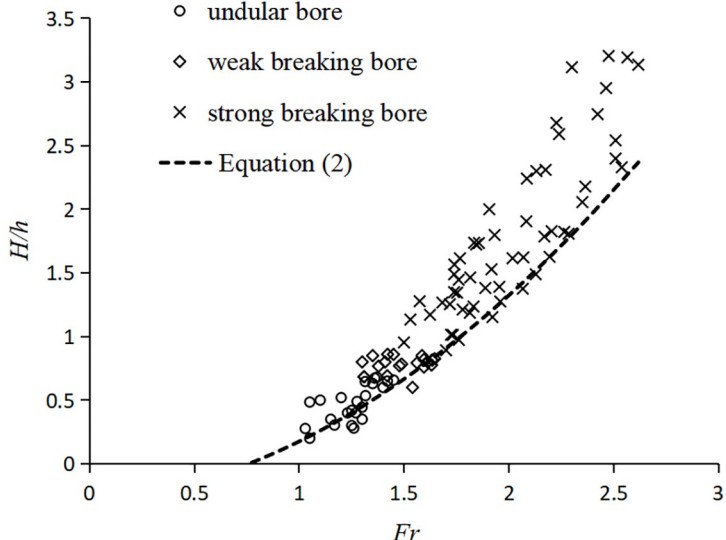

**Fig 8. Forms of tidal bores with *Fr* and *H/h*.**

In a horizontal rectangular channel, the relationship between $H/h$ and $Fr$ for a hydraulic jump is [3]:

$$H/h = \eta - 1 = 0.5(\sqrt{1 + 8Fr^2} - 1) - 1 \qquad (2)$$

where $\eta = (H + h)/h$ is the ratio of hydraulic jump conjugate depth. Here, $H/h$ and $Fr$ for hydraulic jumps are given in the same mathematical format as that of tidal bores. Eq (2) is also depicted in Fig 8. It is noteworthy that the curve for hydraulic jumps is an outer envelope of tidal bores. It is seemly like that tidal bore is similar to moving hydraulic jump.

## Pressure of tidal bore

When $h = 0.096 m$ and $v = 0.2 m/s$, 11 tidal bores with heights from 0.045 m to 0.181 m were measured (Table 1). Notably, one undular bore (case A1), two weak breaking bores(cases B1 and B2) and eight strong breaking bores (cases C1~C8) were observed. The water level process line collected by the sensor was labeled with the height of tidal bore. The pressure process line collected by the sensor was still labeled with the sensor's label. Figs 9–11 show water surface elevation and pressure measured at different sensor heights in the flow. Here the units of pressure have been converted from pascals ($Pa$) to meters of water column($mH_2O$) by dividing by $\rho g$.

As shown in Fig 9, the instantaneous pressure of the undular bore fronts is undulating, and the fluctuation amplitude decreases with the increase of water depth. The period of pressure fluctuation is consistent with the water level fluctuation. Fig 10 shows that the instantaneous pressure of weak breaking bore fronts below ebb flow level (T1~T3) is essentially consistent with its water level fluctuation and increases proportionally with water depth. The amplitude of the fluctuation does not significantly differ from that of the tidal level. The instantaneous pressure of weak breaking bore fronts above ebb flow level (T4~T5) exhibit an immediate increase to a maximum value, followed by a rapid decline and accompanied by subsequent fluctuation. Fig 11 shows that instantaneous pressure of strong breaking bore fronts above the ebb flow level (T4~T7) rose to its maximum value before rapid decline. The amplitude of the subsequent fluctuation deviated from the tidal level and exhibited an increasing trend with rising height. Simultaneously, as the height of tidal bore increased, the maximum instantaneous pressure at a given location also rose in tandem with heightened pressure fluctuations.

Through analyzing the instantaneous pressure change of three forms of tidal bores along the vertical direction, it was observed that the pressure fluctuation of the breaking tidal bore above the ebb flow level exhibits characteristics of impact pressure, while that below the ebb flow level is characterized by pressure fluctuations caused by turbulence. Following the impact

**Table 1. Experimental cases of tidal bore pressure test when $h = 0.096$ $m$ and $v = 0.2$ $m/s$.**

| Case | $H$ (m) | $U$ (m/s) | $Fr$ | $H/h$ | forms of tidal bore |
|------|---------|-----------|------|-------|---------------------|
| A | 0.045 | 1.116 | 1.357 | 0.469 | Undular bore |
| B1 | 0.067 | 1.176 | 1.419 | 0.702 | Weak breaking bore |
| B2 | 0.073 | 1.250 | 1.495 | 0.763 | Weak breaking bore |
| C1 | 0.094 | 1.450 | 1.701 | 0.981 | Strong breaking bore |
| C2 | 0.110 | 1.490 | 1.742 | 1.142 | Strong breaking bore |
| C3 | 0.117 | 1.530 | 1.784 | 1.220 | Strong breaking bore |
| C4 | 0.122 | 1.580 | 1.835 | 1.275 | Strong breaking bore |
| C5 | 0.145 | 1.620 | 1.876 | 1.507 | Strong breaking bore |
| C6 | 0.154 | 1.667 | 1.925 | 1.604 | Strong breaking bore |
| C7 | 0.162 | 1.754 | 2.015 | 1.688 | Strong breaking bore |
| C8 | 0.176 | 1.807 | 2.069 | 1.833 | Strong breaking bore |

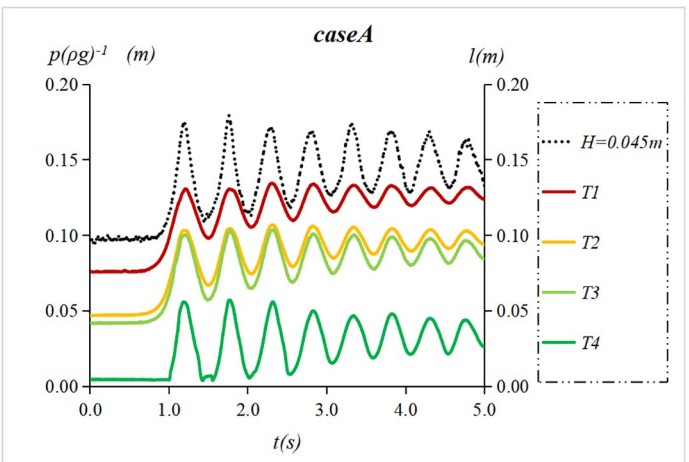

**Fig 9. Undular bores' water surface elevation and pressure when $h$ = 0.096 m and $v$ = 0.2 $m/s$.**

pressure, the pressure fluctuation of the tidal bore above the low water level also demonstrates pressure from turbulence. Therefore, the tidal fronts pressure exhibits dual features of impact and turbulence pressures. The subsequent section will primarily focus on the presentation of impact pressure of the strong breaking bore above the ebb flow level.

## Impact pressure of tidal bore

As analyzed previously, the pressure of the strong breaking bore experiences an instantaneous increase to its maximum value, followed by a rapid decline. This phenomenon is attributed to the strong impact of the tidal bore fronts. At a given height within the tidal bore fronts (red point in Fig 12a), the impact pressure is defined as the maximum instantaneous pressure $p_{max}$ minus the temporally- averaged pressure $\overline{p}$ (Fig 12b), that is:

$$p_i = p_{\max} - \overline{p} \tag{3}$$

Relative to the height of tidal bore, the dimensionless elevation of the tidal bore fronts from surface of ebb flow at a given height within the tidal bore fronts is defined as:

$$Z = \frac{z}{H}, 0 \le z \le H \tag{4}$$

where $z$ is the elevation from water surface of ebb flow at one arbitrary point of the fronts.

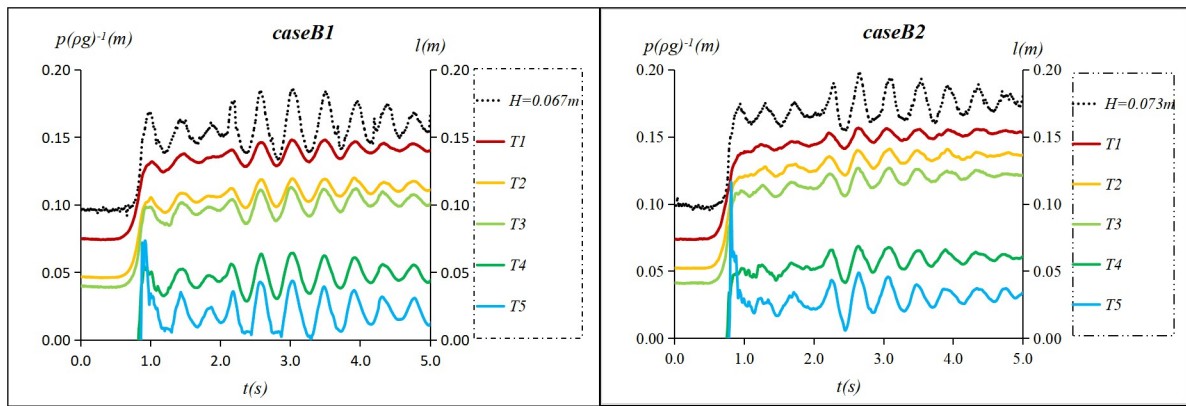

**Fig 10. Weak breaking bores' water surface elevation and pressure when $h$ = 0.096 m and $v$ = 0.2 $m/s$.**

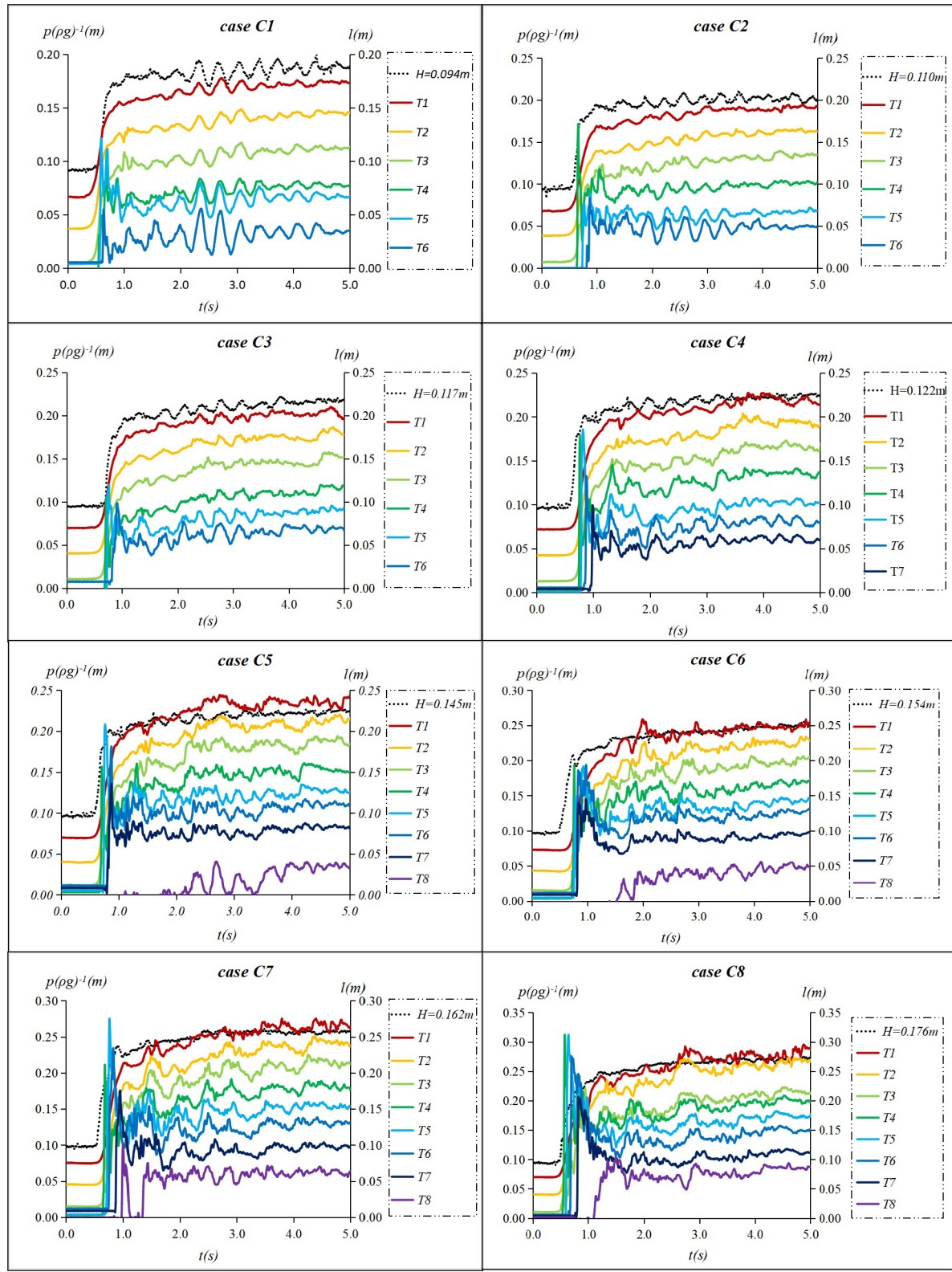

**Fig 11. Strong breaking bores' water surface elevation and pressure when *h* = 0.096 m and *v* = 0.2 *m/s*.**

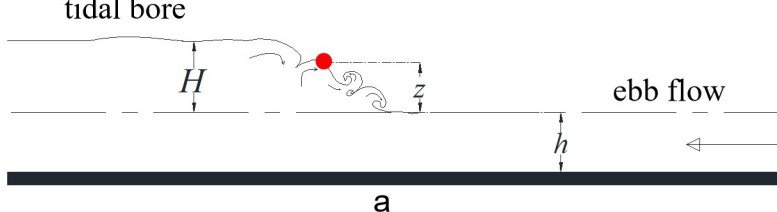

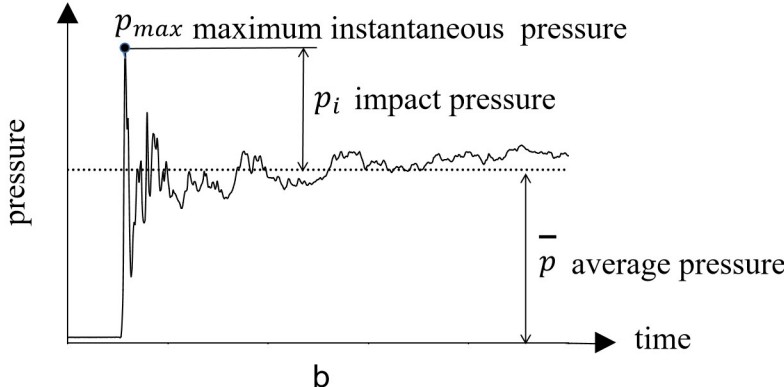

**Fig 12. Schematic diagram of impact pressure of the tidal bore fronts.**

A dimensionless relative impact pressure $P$ at elevation $z$ is defined as:

$$P = \frac{p_i}{\rho g H} \tag{5}$$

where $\rho$ is the density of water, $g$ is the acceleration of gravity and $H$ is the height of tidal bore.

By plotting the non-dimensional $P$ against $Z$, the curve shown in Fig 13 was obtained. It illustrates that the relative impact pressure under different conditions increases rapidly with the increase of the relative elevation of the tidal bore, and then decreases. Simultaneously, the maximum value of the impact pressure is distributed around 0.4 ~ 0.5 times height of tidal bore.

It is also shown that the distribution of $P$ and $Z$ can be described by the probability density function of normal distribution, namely:

$$P = ae^{-\frac{(Z-c)^2}{2b^2}} \tag{6}$$

where $a, b$ and $c$ are constant values.

Through multivariate nonlinear regression analysis data fitting, we determine that $a = 0.807$, $b = 0.243$, and $c = 0.462$. Therefore, the impact pressure of tidal bore can be calculated by the following equation:

$$p_i = \rho g H P = 0.807 \rho g H e^{-\frac{(\frac{z}{H}-0.462)^2}{0.118}} \tag{7}$$

Eq (7) shows that the maximum impact pressure caused by the tidal fronts is at 0.462 times of the tidal height. Substituting Eq (7) into Eq (3), the maximum instantaneous pressure value

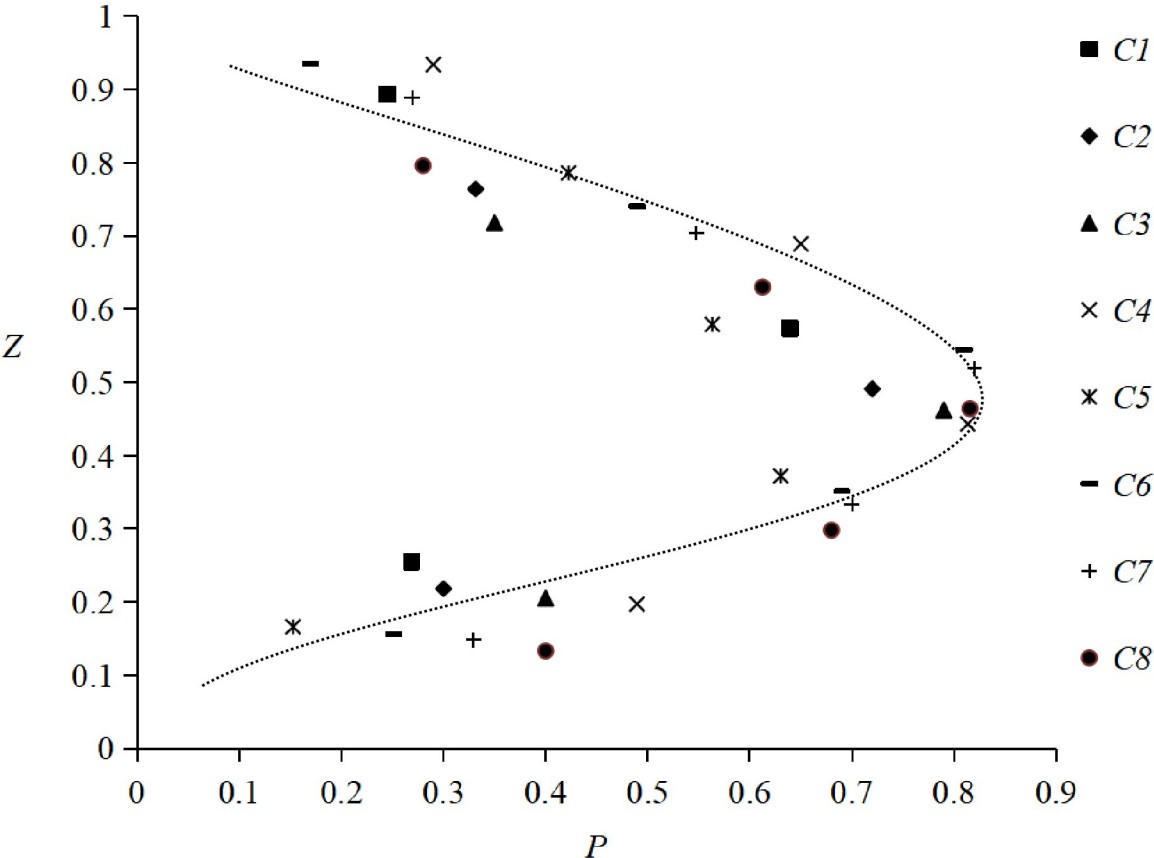

**Fig 13. Relationship of dimensionless impact pressure and elevation of tidal bore fronts.**

of tidal fronts elevation can also be obtained:

$$p_{max} = 0.807\rho gHe^{-\frac{(\frac{\bar{z}}{H}-0.462)^2}{0.118}}+\bar{p} \tag{8}$$

The average pressure value of the tidal fronts is approximately equal to the hydro static pressure at elevation $z$. That is:

$$\bar{p} = \rho g(H - z) \tag{9}$$

Obviously, the maximum impact pressure value of tidal fronts elevation can be calculated by

$$p_{max} = 0.807\rho gHe^{-\frac{(\frac{\bar{z}}{H}-0.462)^2}{0.118}}+\rho g(H - z) \tag{10}$$

Letting $\frac{z}{H} = 0.462$, the maximum instantaneous pressure of the entire tide fronts is:

$$Max[p_{max}] = 0.807\rho gH+\rho g(H - 0.462H) = 1.345\rho gH \tag{11}$$

Field measurements on the pressure were conducted on a cylinder pile at Daquekou in the Qiantang River, China. The Field arrangement of pressure sensors is depicted in Fig 14. Maximum instantaneous pressure of tidal bores extracted from field measurement and calculated from Eq (11) were plotted in Fig 15.

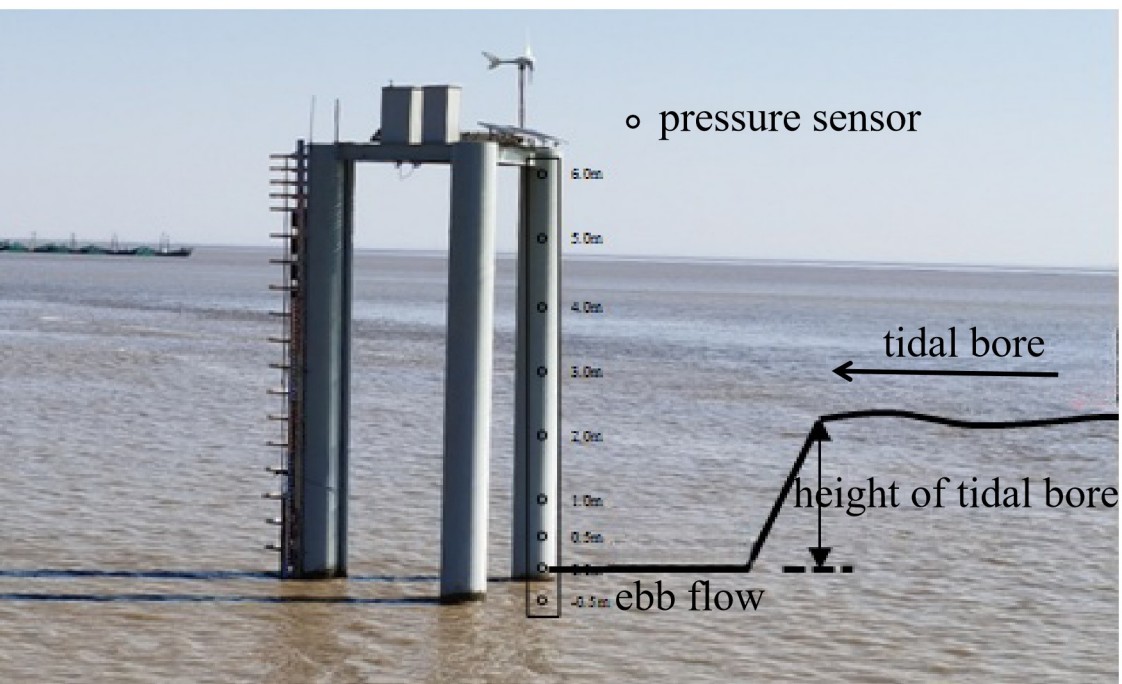

**Fig 14. Field arrangement of pressure sensors at Daquekou in Qiantang estuary, China.**

Previously, Li et al. [26] established a fitting relationship between the pressures extremum and the height of the tidal bore as follows:

$$Max[p_{max}] = 17.55H^{1.35} \qquad (12)$$

Eq (12) also is added to the Fig 15. By comparison, it is found that the calculated value of Eq (12) is larger than the field measured values and experimental values of Eq (11) when the

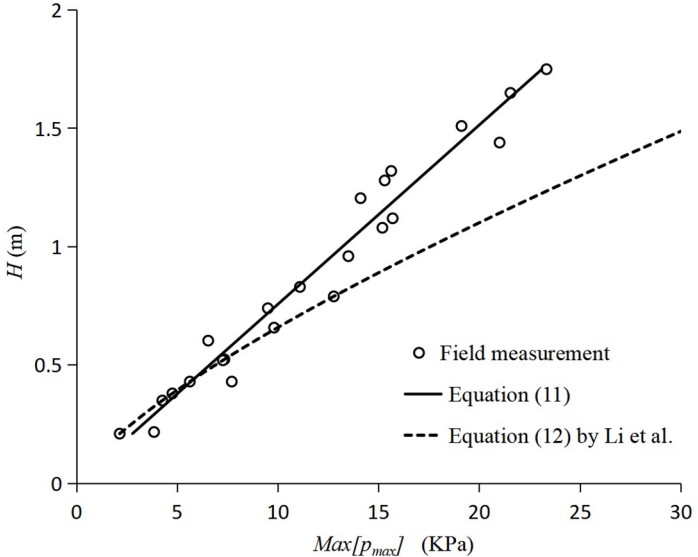

**Fig 15. Values comparison between calculation by Eqs (11) and (12) and field measurement.**

height is larger than 1m. That is most likely because Eq (12) was fitted by the outer envelope of extreme values of the field measurement.

By comparing the field-measured values with the experimental values obtained from Eq (11), it is evident that they are in agreement with each other, and the errors are minimal. Consequently, Eq (11) can be utilized as an empirical equation to calculate the maximum instantaneous pressure of the tidal fronts. Additionally, Eq (10) can also be employed to calculate the maximum impact pressure at different tidal heights. Therefore, both Eqs (10) and (11) can be applied to predict the impact pressure and instantaneous maximum pressure of the tidal fronts, providing essential data support for estuary engineering design and construction.

## Conclusions

In this paper, through physical modelling simulation of tidal bores and analysis of instantaneous pressure, the following conclusions can be drawn:

The amplitude and period of undular bore pressure closely follow the wavy pattern of its water level, with the amplitude decreasing as the water depth increases. Below the level of ebb flow, the amplitude and period of weak and strong breaking bore fronts are essentially in line with its water level turbulent fluctuation. Above the ebb flow level, the amplitude of weak and strong breaking bore fronts instantaneously increases to a maximum value before rapid declining, but the amplitude of the subsequent pressure no longer corresponds to the water level fluctuation. That is the pressure of weak and strong breaking bore fronts exhibits dual features of impact and turbulence. The maximum impact pressure of strong breaking bore appears around 0.46 times the tidal bore height. Through non-dimensionalization of impact pressure and elevation of tidal bore fronts, they conform to the normal probability density function. Equations for calculating impact pressure and maximum instantaneous impact pressure are obtained through multiple linear regression analysis.

## Supporting information

**S1 Dataset. The raw data is used to plot Fig 7.**
(XLSX)

**S2 Dataset. The raw data is used to plot Fig 8.**
(XLSX)

**S3 Dataset. The raw data is used to plot Fig 9.**
(XLSX)

**S4 Dataset. The raw data is used to plot Fig 10.**
(XLSX)

**S5 Dataset. The raw data is used to plot Fig 11.**
(XLSX)

**S6 Dataset. The raw data is used to plot Fig 13.**
(XLSX)

**S7 Dataset. The raw data is used to plot Fig 15.**
(XLSX)

## Author Contributions

**Conceptualization:** Shubo Yue.

**Formal analysis:** Shubo Yue, Zhiyong Zhang.

**Investigation:** Shubo Yue, Zhiyong Zhang.

**Methodology:** Shubo Yue, Jian Zeng, Yongping Chen.

**Validation:** Shubo Yue, Jian Zeng.

**Writing – original draft:** Shubo Yue.

**Writing – review & editing:** Shubo Yue, Dongfeng Xie.

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
