## [Decision Letter · Decision Letter 0]

15 Nov 2023

PONE-D-23-21028Experimental research on the impact pressure intensity of tidal bore frontPLOS ONE

Dear Dr. YUE,

Thank you for submitting your manuscript to PLOS ONE. After careful consideration, we feel that it has merit but does not fully meet PLOS ONE’s publication criteria as it currently stands. Therefore, we invite you to submit a revised version of the manuscript that addresses the points raised during the review process.

Overall, the structure of the paper is good and will attract the reader. However, the following comments should be addressed, in addition to the reviewer comments, before considering the article for publication.

The language in the article contains numerous grammatical mistakes. These issues significantly hinder the reader's comprehension and should be rectified to enhance the manuscript's overall quality.List of nomenclature should be included and each abbreviation should be explained.Introduction section is weak and the authors missed the latest relevant research. Most of the literature is outdated. Author should discuss the finding of the earlier research and also discuss the limitations of those studies. The literature study can be enhanced by adding recent relevant references (last five years).All the variables should be explained in paper context.

We look forward to receiving your revised manuscript.

Kind regards,

Niaz Bahadur Khan, PhD

Academic Editor

PLOS ONE

Journal Requirements:

"This research was funded by National Nature Science Foundation of China , grant number 41376099 and 51609214 and Zhejiang Provincial Natural Science Foundation of China, grant number LY20E090001."

"This research was funded by National Nature Science Foundation of China , grant number 41376099 and 51609214 and Zhejiang Provincial Natural Science Foundation of China, grant number LY20E090001. "

"This research was funded by National Nature Science Foundation of China , grant number 41376099 and 51609214 and Zhejiang Provincial Natural Science Foundation of China, grant number LY20E090001. "

7. We note that Figure 1 in your submission contain copyrighted images. All PLOS content is published under the Creative Commons Attribution License (CC BY 4.0), which means that the manuscript, images, and Supporting Information files will be freely available online, and any third party is permitted to access, download, copy, distribute, and use these materials in any way, even commercially, with proper attribution. For more information, see our copyright guidelines: http://journals.plos.org/plosone/s/licenses-and-copyright.

Additional Editor Comments:

Overall, the structure of the paper is good and will attract the reader. However, the following comments should be addressed, in addition to the reviewer comments, before considering the article for publication.

• The language in the article contains numerous grammatical mistakes. These issues significantly hinder the reader's comprehension and should be rectified to enhance the manuscript's overall quality.

• List of nomenclature should be included and each abbreviation should be explained.

• Introduction section is weak and the authors missed the latest relevant research. Most of the literature is outdated. Author should discuss the finding of the earlier research and also discuss the limitations of those studies. The literature study can be enhanced by adding recent relevant references (last five years).

• All the variables should be explained in paper context.

Reviewers' comments:

Reviewer's Responses to Questions

**Comments to the Author**

1. Is the manuscript technically sound, and do the data support the conclusions?

Reviewer #1: Yes

Reviewer #2: Yes

Reviewer #3: Partly

2. Has the statistical analysis been performed appropriately and rigorously? 

Reviewer #1: Yes

Reviewer #2: Yes

Reviewer #3: Yes

3. Have the authors made all data underlying the findings in their manuscript fully available?

Reviewer #1: Yes

Reviewer #2: Yes

Reviewer #3: No

4. Is the manuscript presented in an intelligible fashion and written in standard English?

Reviewer #1: Yes

Reviewer #2: No

Reviewer #3: No

5. Review Comments to the Author

Reviewer #1: In this paper, the influence factors of the tadial front impact pressure are explored through flume tests, and the calculation formula of the impact pressure is fitted. Finally, the validity of the calculation formula is verified by the measured data of the cylinder piles of Qiantang River. The research method is reliable, and the research conclusion has certain significance in engineering practice. However, the following questions remain.

(1) The flow velocity sensor should also be installed in the test, otherwise the flow velocity cannot be obtained, please add the corresponding description in the test setting.

(2) The h and v used in lines 81 and 82 do not seem to have been defined before using them.

(3) Please confirm whether there is any problem with the statement in line 116. This statement looks like it needs polishing.

(4) What exactly do the "11 values" in line 71 refer to?

(5) The resolution of fig. 1, 3, 5, 6, 7, 8, 11, 12 is too low, please use a higher resolution picture if possible.

(6) Fig. 2 Please add the necessary graphic illustration.

(7) Redraw Fig. 4, use different colors, or linear lines to distinguish each line, and add legends.

(8) In line 90, please use the correct italics for h and v, and check the full paper.

(9) The full text paragraph format is not uniform, such as 121 lines, please unify the format.

In summary, it is suggested to make a decision after the major revision of this paper.

Reviewer #2: This manuscript extensively studied the impact pressure intensity of the tidal bore front. A physical model was built to simulate the tidal bores in a glass rectangular flume. Then, the author studied the tidal bores from three perspectives: 1) the forms of the tidal bores; 2) the pressure intensity; 3) the impact pressure. In all, the study is interesting. Please consider the following problems:1) This manuscript contains massive writing issues and should be carefully corrected and improved. Please proofread the whole manuscript and make corrections. 2) The meaning of each letter in mathematical equations should be explained. Please make corrections.3) Mathematical, by replacing Eq.(5) with Eq. (4), the result is not a normal distribution (Eq.6). Please give more details of the derivation of Eq.(6). Please clarify.4) Eq.(11) is a linear equation, and Eq. (10) is a nonlinear equation. A problem arises: why Eq. (10) can be approximated by Eq. (11). Please clarify. 5) The coefficients in Eq.(7) ~Eq.(11) are constant, can they be generalized to the resulting equation and made a derivation?6)  The conclusions are ok, and I do not have additional problems.7)  references:  (1) the surname from Ref.1~Ref.14 is incomplete. (2) Ref. 4 and Ref. 6 should add the page number.

Reviewer #3: This review is of “Experimental research on the impact pressure intensity of tidal bore fronts”. Overall I think the results are interesting but I have suggestions for substantial improvement and clarification of motivations and methods. I hope my suggestions are taken constructively and result in an improved manuscript.

I think the problem the authors are studying—impact intensities from hydraulic bores—is much more relevant and broadly applicable than the authors make it out to be. The topic is very relevant to flash floods, tsunamis, dam break floods, and rogue waves in coastal areas. As the authors themselves say, tidal bores are “limited to only a few locations worldwide”. This is not the way to convince readers of the importance of your research! I recommend emphasizing that the experimental results are applicable to a range of natural phenomena that are important to people and coastal environments, and only then focusing on the narrow case of tidal bores.

I recommend adding a simple schematic diagram of the flume instrumentation, including where measurements were done, and that also includes a diagram of a tidal bore showing definitions of variables. For many variables I was not sure where they were measured to or from, and I had to guess.

Finally, I think more details on the pressure measurements are needed. What type of sensor was used? Is it measuring impact forces? Pore pressure? In particular, does it work instantaneously when going from dry to wet? Many pore pressure transduces that I’m familiar with would give numbers, but not really meaningful data, when going from dry to submerged. How quickly do the sensors themselves respond, and what is the sampling rate of recorded data? It is possible that the instrumentation used is appropriate for measuring from dry to wet conditions with impact forces coming from one direction, but the authors should demonstrate that their data can do these things, starting with describing the type of pressure sensors used in the experiments.

Title: I’m not sure the word “intensity” should be in the title, or used in exactly this way. I realize that “intensity” can be used informally or in different ways, but often in physics it is a measure of power, or energy per time. I suggest “Experimental research on impact pressures of tidal bore fronts” or something similar. The authors should consider changing the usage of intensity in other places in the manuscript as well.

Page numbers would be helpful.

Line 8: suggest changing the English to “Tidal bore impacts can be sufficiently strong and destructive…”

Lines 33-34, and other places in this and the next paragraph: Don’t just say that a given author studied a topic, say what they found. Descriptions of previous work should give insight or knowledge into the process being studied, not just say that people studied it.

33: what is a “model test”? Laboratory experiments? Numerical modeling?

45: cut “preliminarily”; if its published it should no longer be a preliminary result.

66 and below: I think a figure showing from where to where water depth and tidal height are measured is needed, and which direction is considered positive for ebb velocity. How is water depth different from the height of the tidal bore front H? Also, where is z measured (eqn5)?

70, 71: Variables appear to not show up in the pdf I downloaded; I assume there is supposed to be a variable in “(represented by )”.

Figure 2: Its odd to have a figure that mostly shows side-looking ADVs, and then never mention them or saying that you collected data with them but don’t present it here. Is the other device the capacitance-based wave gauge? Consider replacing with a schematic of the pipe containing the pressure probes.

80: Change “front” to “fronts”.

83: I’m not sure what “It is show that the break of the free surface of tidal bore” is trying to say.

86: Odd to call a relative height of the tidal bore H/h a “strength”. Suggest calling it a relative height or depth. I’m not sure where either H or h are measured without a diagram.

94-95: These two sentences basically say the same thing; change to something like “Figure 5 shows that the three classes of tidal bores vary systematically with Fr and H/h”.

102: Say what aspect of the results are in accord with field data. How are they consistent? What did Xie find?

Equation 2: Give a citation for this equation; if the authors came up with it themselves, make that more clear from the text. I think this is expected for a hydraulic jump.

114: change tidal to tidal bores

116: change “unit of intensity of pressure” to just “units of pressure”

Figure 6, also relevant for 7 and 8: I don’t think T1, T2, etc. are defined anywhere. I assume these are different pressure sensors; make that clear and say what heights they were measured at. I can guess that T1 is lowest height and T4 is highest, but just give the reader that information. Also, I think but am not certain that the y axes “l/m” is not saying l divided by m, or per meter, but that “l” in units of meters? Change the notation on all the plots to not use the slash (“/”) to indicate the units, because it usually means divide by. Also, was l ever defined? Is it different from h? Again, a diagram defining variables is needed, as is consistency. Also, I don’t think “t” was ever defined, though I assume the x axis is time in seconds?

121-122: This seems to say that the instantaneous pressures are just hydrostatic (increase proportionally with water depth), which seems to be contradicted by the next sentence (123-124) in which pressures increase then decrease as one moves up from the bed. Which is it?

137: what is the backward fluctuation? I don’t know what this means.

142: what is “pressure pulsation”? I don’t think this was defined.

171: Consider giving regression confidence intervals on the parameters in the normal distribution pdf (a, b, c). Change the English from “it can be get that” to “we determine that” or similar.

174: 0.462 is too precise; suggest just saying 0.5. Its odd to say earlier that the maximum values are between 0.3 and 0.6 (a big range), and then be so specific (0.462) here.

If there were supplemental materials containing data or other descriptions I could not find them.

6. PLOS authors have the option to publish the peer review history of their article (what does this mean?). If published, this will include your full peer review and any attached files.

Reviewer #1: No

Reviewer #2: No

Reviewer #3: No

---

## [Author Response · Author response to Decision Letter 0]

27 Jan 2024

We are grateful for the comments of three reviewers and editor, which have greatly helped us to improve the quality of this manuscript. Our responses to the comments are given in the File: Resposes to reviewer's comments.

---

## [Decision Letter · Decision Letter 1]

13 Mar 2024

Experimental research on the impact pressure of tidal bore fronts

PONE-D-23-21028R1

Dear Dr. YUE,

We’re pleased to inform you that your manuscript has been judged scientifically suitable for publication and will be formally accepted for publication once it meets all outstanding technical requirements.

Kind regards,

Niaz Bahadur Khan, PhD

Academic Editor

PLOS ONE

Additional Editor Comments (optional):

Authors have addressed all the reviewers' comments, and the revised manuscript now significantly improves upon the clarity and depth of the research. However, upon careful review, I have observed some minor grammatical mistakes that still need attention.

Reviewers' comments:

Reviewer's Responses to Questions

**Comments to the Author**

1. If the authors have adequately addressed your comments raised in a previous round of review and you feel that this manuscript is now acceptable for publication, you may indicate that here to bypass the “Comments to the Author” section, enter your conflict of interest statement in the “Confidential to Editor” section, and submit your "Accept" recommendation.

Reviewer #1: All comments have been addressed

Reviewer #3: All comments have been addressed

2. Is the manuscript technically sound, and do the data support the conclusions?

Reviewer #1: (No Response)

Reviewer #3: Yes

3. Has the statistical analysis been performed appropriately and rigorously? 

Reviewer #1: (No Response)

Reviewer #3: Yes

4. Have the authors made all data underlying the findings in their manuscript fully available?

Reviewer #1: (No Response)

Reviewer #3: Yes

5. Is the manuscript presented in an intelligible fashion and written in standard English?

Reviewer #1: (No Response)

Reviewer #3: Yes

6. Review Comments to the Author

Reviewer #1: (No Response)

Reviewer #3: The manuscript by Yue et al. is greatly improved. I appreciate the care and thoroughness they showed in their revisions and their responses to reviewers. I have various minor suggestions for improvement, and recommend its publication after minor revisions. The English usage is also greatly improved; it is still a little rough in places but understandable throughout. This is a nice paper.

I think the additional figures and schematics really help. The figures in the pdf I read were still fuzzy and low resolution.

Figure 2: I don’t know if it is a resolution issue, but I can’t figure out where the ebb flow goes, or how the tidal bore gets into the flume. It looks like there is a vertical wall (that then overhangs part of the flume) separating these sections. My guess is that there is not actually a vertical wall there, and that the ebb flow goes into the water recycling tank until the gate is dropped? But I’m not sure. I suggest changing the diagram to get rid of that wall, and show where the ebb flow goes before the bore is released (if I understand how the flume works correctly.

The authors should consider adding before-bore release and after-bore release panels to make it more clear. At the moment the figure shows the bore in the flume, even before the gate is dropped. Or, I don’t know if this idea would be better or just too complicated, but the figure could be drawn using different colors to reflect the water and gate both before and after the gate is dropped.

Beyond that most of my comments are to improve English.

13: suggest changing slightly to “Tidal bore impacts can be strong and destructive, placing estuarine infrastructures under great threat.”

16-17: after reading the whole manuscript I see that the authors are trying to say that impact pressures are what matter to the overall pressures exerted by tidal bores. But when I first read it I didn’t realize that “impact” was the key point, and it sounded like the sentence just said that pressures depended on pressures. Suggest rewording to make even more clear that the distinction is from the impact of a tidal bore.

30: Change “pose” to “cause”.

34-35: suggest changing to “…investigated how the water surface slope of tidal bore fronts varied with changes in estuary channel morphology. They found that decreases in channel width and depth promoted the generation of tidal bores.”

44: Madsen and Svendsen are the only two authors (as given in the references), remove the “et al.”

62: Change It’s to It is

69: Change “dynamics measurement” to “measurements”

83-84: suggest changing to “However, equations describing how pressures caused by tidal bore impacts have not been presented.” (if true; it seems like many of the previous citations did this? With the parabolic vs triangular findings?

108: change “So H could be calculated by” to “H was calculated by”. Also change “could be” to “was” at the end of the sentence.

150: Change “cited” to “given in”

156: Change “are observed” to “were observed”

156-162: I am not sure what “level process line” and “pressure process line” mean; this is odd usage. Do you just mean the lines on Figures 9 to 11 that show water level and pressure? Just say something like “Figures 9, 10 and 11 show water surface elevation and pressure measured at different sensor heights in the flow”.

177, 178: Change “pressure turbulence” to “pressure fluctuations caused by turbulence” in the first usage, and “pressure from turbulence” in the second.

183-184: Change “ At one arbitrary position of tidal bore fronts” to “At a given height within the tidal bore front “

185: change “average pressure” to “temporally-averaged pressure”.

196: Change “a curve shown in Figure 13 can be obtained” to “the curve shown in Figure 13 was obtained”.

219: Should be figure 15, not 16.

228: Figure 15 caption: change “Equation(11)(12)” to Equations (11) and (12).

7. PLOS authors have the option to publish the peer review history of their article (what does this mean?). If published, this will include your full peer review and any attached files.

Reviewer #1: No

Reviewer #3: No
